# TYPHABOARD in the Restoration of Historic Black Sea Houses in Bulgaria

**Georgi Georgiev [1,\*], Martin Krus [1], Carina Loretz [1] and Werner Theuerkorn [2]**

[1]    Fraunhofer-Institute for Building Physics, Fraunhoferstr. 10, 83626 Valley, Germany;
        martin.krus@ibp.fraunhofer.de (M.K.); carina.loretz@ibp.fraunhofer.de (C.L.)

[2]    Werner Theuerkorn, Wichtleiten 3, 84389 Postmünster, Germany; w.theuerkorn@gmail.com

**\***    Correspondence: georgi.georgiev@ibp.fraunhofer.de; Tel.: +49-8024643612

**Abstract:** TYPHABOARD is a sustainable construction and thermal insulation board made of cattail (lat. typha) and magnesite as a binder. It is characterized by a unique combination of high mechanical strength, highly insulating properties, relatively high diffusion openness, inflammability, and a sustainable lifecycle. The TYPHABOARD concept includes ecological benefits related to the systematic planting of the raw material typha in Bulgaria, the production of TYPHABOARD, and its application as a stabilizing, insulating and passive indoor climate controlling element in the framework structure of the historic typology of the Black Sea House. The entire technological and organizational process provides a sustainable solution for the operation of peat areas by planting typha (which acts as a natural water and ground filter), for the engagement of work forces in structurally underdeveloped regions, for sustainable ecological and social regional development, as well as for the sustainable retrofit of existing historic Black Sea Houses. The building and ecological system TYPHABOARD can be successfully introduced and applied in Bulgaria. In addition to the scientific and the practical study, the political eligibility of this was proved and officially permitted by the relevant public bodies in Bulgaria.

**Keywords:** Typha; cattail; TYPHABOARD; building material; indoor insulation; half-timber wooden structure; Black Sea House; historic buildings restoration; Bulgaria

## 1. Introduction

Through the development of TYPHABOARD, a natural, sustainable and recyclable material for the building industry has been developed. Besides its positive environmental effects, it also has good insulation properties as well as good mechanical properties. TYPHABOARD represents an innovative and promising construction material in accordance with the natural carbon lifecycle. As has already been proved, the board offers great potential for the monument-related restoration of historic buildings with framework structures [1]. A prospective application for TYPHABOARD is the restoration of Black Sea Houses, a historic, half-timbered building typology prevailing along the Black Sea Coast in Bulgaria.

Within the framework of the TYPHABOARD concept, three main projects have already been conducted under support of the Deutsche Bundesstiftung Umwelt DBU, the German Federal Environmental Foundation. The first project (AZ 10628), in the period from 1998 to 2001, focused on an exploration of the sustainable cultivation of typha in peat and moor areas in Germany and was conducted by the Technical University of Munich. Its final report was published under the German title "Rohrkolbenanbau in Niedermooren—Integration von Rohstoffgewinnung, Wasserreinigung zu einem nachhaltigen Nutzungskonzept" [2]. Another model project (AZ 27918), with the title "New Building Material for Environmentally Compatible and Structural Retrofitting in Monument Preservation", was

conducted from 2010 to 2013 to test the application of TYPHABOARD as infill material and indoor insulation for the renovation of a half-timbered, medieval craftsman building in Nuremberg. Within this project, TYPHABOARD manufacturing technology was developed and the boards were installed as infill for damaged Fachwerk-fields, as well as indoor insulation at the Fachwerk façade of the pilot Fachwerk house [3]. The most recent project (AZ 31996) on the TYPHABOARD concept, which ran from 2015 to 2017, investigated the implementation of TYPHABOARD as indoor insulation at the energy retrofit of multi-occupancy residential buildings with owner communities in Bulgaria. Its German title is "Prüfung der Übertragbarkeit eines neu entwickelten, innovativen, nachhaltigen Baustoffs zur thermischen Sanierung von Mehrfamilienhäusern mit Kleineigentümerstruktur in Bulgarien". Due to the high number of multiple owner-occupied residential buildings in Bulgaria and the difficulty involved in reaching an agreement among the owner community on deep renovation, energy optimization of entire residential buildings is mostly not possible. Internal insulation in individual apartments seems to be a promising alternative. The project revealed that the TYPHABOARD concept is well suited for implementation in Bulgaria and involves several socio-economic and ecological advantages. The results of the study indicated that the highest potential for TYPHABOARD application, among others, lies in the monument-related restoration of historic buildings with a wooden, and in particular, half-timber framed structure [4,5]. One example is the typology of the Black Sea House, a particular locally specific building type from the last five to six centuries, which occurs in regional areas around the Black Sea. Most of the available information about this building typology is supplied by Professor Tuleschkow through oral transmission and informal text collection, due to the highly craftsmanship character of the topic. Typically, the Black Sea House has a facade, which consists of a half-timber structure and is covered by wooden shingles protecting it from extreme weather conditions in the region—such as humid salty winds and rains—and enabling the drying of the construction by the air layer between it and the bearing structure. Nowadays these historic buildings are often in an urgent need of rehabilitation.

A current goal is to develop a strategy for the monument preservation of Black Sea Houses under the application of TYPHABOARD in consideration of the results from previous projects. In particular, the restoration project of the half-timbered medieval building in Nuremberg serves as an informative basis for this future project.

This paper is about the raw material typha in general, and also about its positive environmental effects and its excellent properties. Furthermore, the building material TYPHABORD is introduced, as well as its implementation concepts. To continue with the main topic of this paper, an analysis of the Black Sea Houses' typology and an insight into their history follows. Special requirements for the restoration of Black Sea Houses are clarified, and experience gained through previous projects is used to identify the potential for applying TYPHABOARD in the sustainable energetic and functional retrofitting of Black Sea Houses.

## 2. Materials and Methods

### 2.1. The TYPHABOARD Concept

#### 2.1.1. Typha Plant

Typha, or cattail, is a reed plant which occurs naturally in swamp areas and on river banks. Its growth starts from spreading rhizomes or from seeds, and the plants typically reach a height of two to three meters, depending on the water level conditions, and in the Danube Delta area they can grow up to five meters. Optimal growth is obtained in constantly submerged areas with a water level of at least 30 to 50 cm [6]. Typha has high growth productivity and therefore it is an appropriate raw material for industrial application. A field of typha plants as a monoculture can produce 15 to 20 tons of dry mass per hectare per year, which can be transformed to approximately 150 to 250 m$^3$ of building material [4]. Typha is characterized by its particular physical structure, which is significant for its suitable properties as a construction material. Due to a complex and stable sustentacular tissue within typha leaves, the

plant shows high performance in compressive and tensile strength as well as in elastic deformability. Despite the high porosity of the typha plant, the particles grow symmetrically and in rod-shapes, resulting in good fissility in the longitudinal direction. Due to its spongy tissue (see Figure 1a), typha leaves have a very low density of approximately 40 kg/m$^3$ [3]. The plant mass consists of about 85% aerenchyma tissue, which comes along with low thermal conductivity ($\lambda$ = 0.032 W/mK) and therefore has excellent insulating properties. Furthermore, typha shows high resistance against mildew and bacteria, because of its high polyphenol content. This makes typha a durable building component as well as a resistant agricultural crop without pesticide or fertilizer treatment [6].

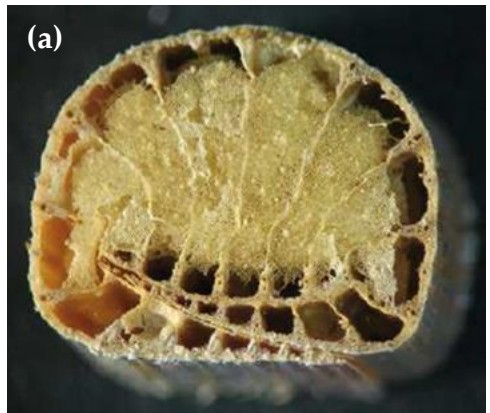 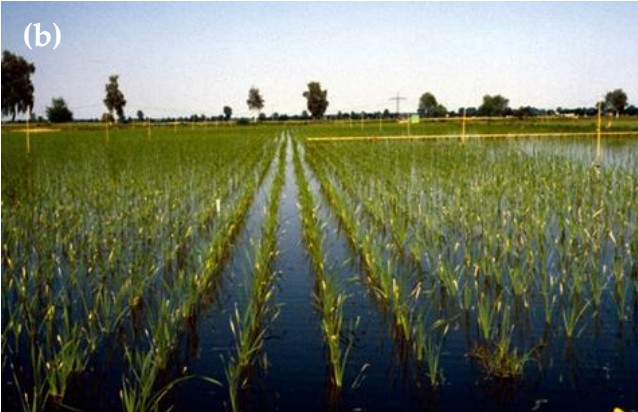

**Figure 1.** (**a**) Section of a typha leaf, picture: Chr. Gruber, BLfD (**b**) typha cultivation, picture: typha technik Naturbaustoffe.

Besides having excellent properties for the building industry, typha cultivation (see Figure 1b) shows a promising potential for solving environmental problems. The plants have the ability to clean polluted water by filtering out harmful substances like nitrates, ammonium, phosphates, copper, nickel and organic pollutants. Furthermore, planting with typha prevents wetlands from drying out and thus it prevents the release of the greenhouse gases carbon dioxide and nitrous oxide, which are retained in natural wetlands. Consequently, the ecological cultivation of moor land is in harmony with nature and of special interest for environmental protection [5,6].

### 2.1.2. TYPHABOARD

The building material TYPHABOARD (see Figure 2) is a 100% natural composite material consisting of precisely chopped particles of typha leaves with magnesite as a binding agent. The boards are formed under relatively low pressure and the production process proves to be simple and therefore relatively cost-effective. The TYPHABOARD acts as insulation as well as construction material, shows high mildew resistance, and can be easily machined and installed [3]. Due to the high insulting effect ($\lambda$ = 0.050 W/mK) and the high load-carrying capacity of about 1 N/mm$^2$, TYPHABOARD can be perfectly used as an indoor insulation panel without great installation effort. A further advantage is the relatively high diffusion value of about 20 that provides protection against condensation loss and offers the possibility of installation beneath wall heating systems [5].

Within the framework of the previous project on the renovation of a medieval craftsman-made building in Nuremberg with TYPHABOARD, the necessary number of boards was produced by the available laboratory equipment. Typha plants as raw material were procured from the Danube Delta in Romania and were delivered in cut and bundled form from Tulcea to Germany. The THYPHABOARD production started with a longitudinal cutter with counter-rotating pairs of knives, which chopped the individual typha leave bundles into bars of 2 to 5 mm. A conveyer transported these to the cross cutter, where the bars were crosscut by a rotational cutter into sections of approximately 10 cm. Transportation to temporary storage followed, where the cut particles were subsequently taken by a conveyer for weighting out in the glue barrel. The material was glued during mixing by a compression unit that

sprayed the solved magnesite particles into the barrel. Subsequently the glued material was fitted into a mold, in this case a 1 × 2.5 m large mold constructed from textured, coated boards with ash wood reinforcing ribs. The material needed to be distributed as equally as possible before the lid was put on the mold. The mold was set into the compactor to compress it to the desired size and then locked in this state. The mold can be taken from the compactor and remain in the drying room, ideally a heated drying room to shorten the time of compression, until the board is cured [3].

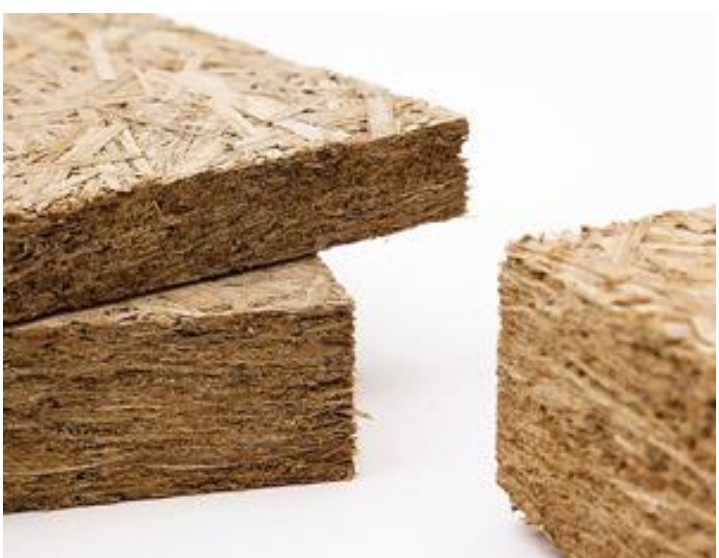

**Figure 2.** TYPHABOARD, picture: Fraunhofer IBP.

### 2.1.3. The TYPHABOARD Concept in Bulgaria

The implementation of the TYPHABOARD concept in Bulgaria besides having positive environmental impacts, is also correlated with beneficial socioeconomic effects. Within the framework of the previous DBU supported project about the implementation of TYPHABOARD at the energy retrofit of multi-occupancy residential buildings in Bulgaria (2015–2017), the prevailing circumstances as well as potential regions for the cultivation of typha plants and for the production of TYPHABOARDS were investigated. Therefore, decisive factors were on the one hand appropriate natural landscape conditions for cultivation, and on the other hand, potential production facilities close to the growing region, sufficient skilled labor on site and a preferably regional market. As the production of TYPHABOARD is based on simple manufacturing processes, there is not any need for a specially qualified workforce. Through the implementation of the TYPHABOARD concept in Bulgaria, new jobs could be created, and the regional economy could be strengthened [4,5].

Within the previous project, potential regions for typha cultivation and TYPHABOARD manufacturing in Bulgaria could be identified: the northeast regions along the Danube, especially the area around Razgrad and Silistra, and the regions around Plovdiv, Stara Zagora, Yambol and Smolyan along the river Maritsa. In particular, Razgrad, Silistra and Yambol (see Figure 3) are located only 100 to 140 km away from the Black Sea Coast. Therefore, these locations may be appropriate sites for regional production facilities in regards to the decentral application in the restoration of Black Sea Houses. [6,7]. Near Bourgas, there are several wetlands that offer the perfect conditions for typha growth, which can be seen nowadays in the region.

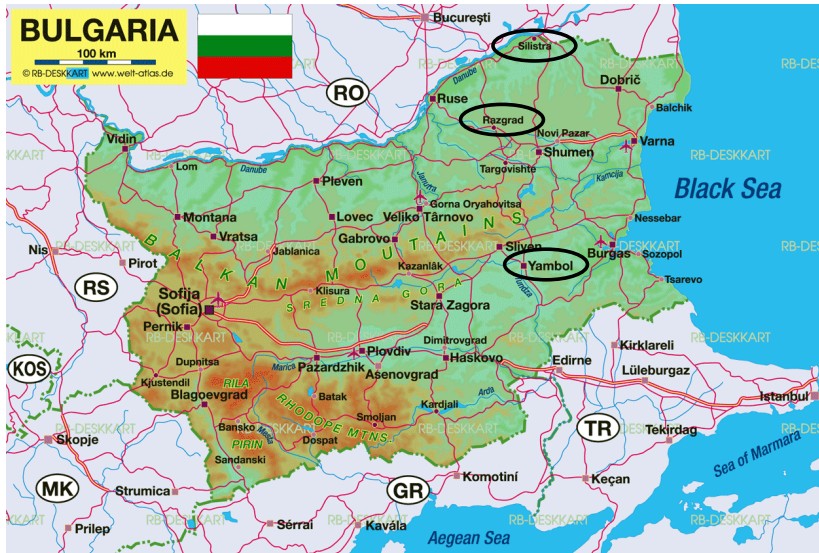

**Figure 3.** Map of Bulgaria, source: www.welt-atlas.de.

*2.2. The Black Sea Houses*

The Black Sea Houses' building typology is a unique regional architectural phenomenon in Europe. Due to the characteristics of its construction system and the previously used construction materials, preserved buildings of this type disappear with time or get renovated without attention to monumental protection.

A main aim of the current article is to present the typology of the Black Sea House and to investigate the opportunity for all examples of this typology to get structurally, functionally and energetically optimized under the application of TYPHABOARD, following a retrofit process scheme standardized for this typology.

2.2.1. Typology of Black Sea Houses

The construction of typical Black Sea Houses (see Figure 4) consists of a classical framework in which the characteristic detail is the facade. The infill of the facade is either made out of clay or adobe or, according to the antique Bulgarian system, with a network of twigs in the middle of the infill in combination with thick clay layers on both sides. On the inside, a thin clay layer is applied to cover the wooden construction, and on it a lime plaster differently colored by natural pigments. The main characteristic though, is the external facade, which is made of vertically or horizontally installed wooden planks. North of Istanbul, one can find facades with second protective layers out of overlapping thin planks, nailed like shingles. Wood paneling proves totally suitable for weather protection in coastal areas where plasters would be destroyed faster by the aggressive environment of strong winds, heavy rainfall, salt, and humidity.

The Black Sea House has its characteristic as well in its functional schematic. Typically, there are large hall-like salons on the ground floor for maintenance work and for making fishing nets, with balconies for drying fish, and several rooms for the occupants around the salon, cellar rooms and restrooms. Houses which were built after the beginning of the 19th century, under the influence of Istanbul, have wooden facade decorations in the style of the empire such as pilasters, cornices and triangle fronts above windows [8].

The architecture of the Black Sea House was mainly implemented by Bulgarian builders, but spread to areas around Istanbul, to eastern regions near the Black Sea, from the northeast of the Middle Sea to Izmir, and from the western White Sea Coast to the delta of Maritsa. Though the largest concentration is around the Black Sea Coast in Bulgaria as well as in Istanbul and in settlements in the surrounding areas [8].

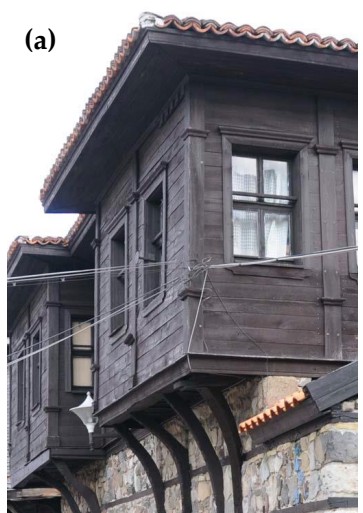
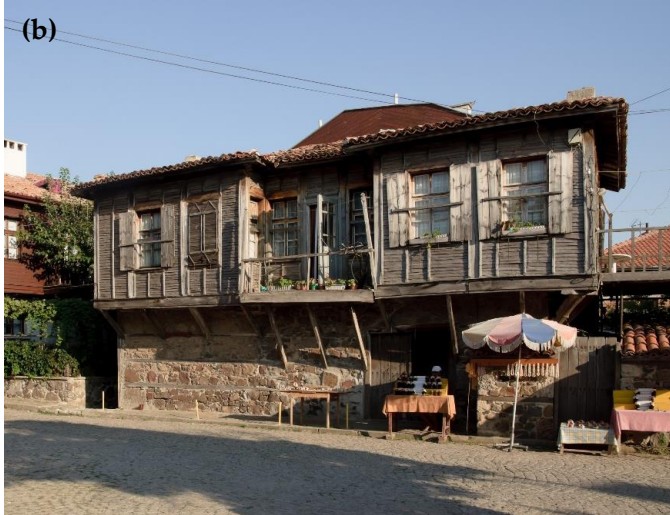

**Figure 4.** Typical Black Sea Houses in Sozopol, Bulgaria (**a**) picture: Nikolay Tuleschkow (**b**) source: www.studiolemontree.com.

### 2.2.2. History of Black Sea Houses

The history of the Black Sea House goes back to the 15th and 16th century. Bulgarian, Armenian and other builders established the basis for this building typology in the surrounding areas of the Bosporus. The urban landscape of old settlements and historic cities around the Black Sea is still characterized by these buildings. In the western regions of Bulgaria and Turkey, Black Sea Houses are well preserved. This building typology acts like a bridge between Bulgaria and Turkey as it indicates their common history and a common origin of architecture.

The main questions about Black Sea House typology are: where the historic roots lie; how far back they are dated; and what transformations they will undergo in the future. Due to a lack of representative data sources, scientific statements are difficult to make. Even though there are a sevral written information sources on the Black sea house typology development, the used data on this topic mainly comes from oral transmission from local construction craftsmen and hobby historians.

The earliest source, which gives an indication of the existence of facade cladding with wooden pieces in the historically almost completely Bulgarian city of Edirne (nowadays Turkish), comes from Caterion Zeno (1550), a Venetian diplomat. He describes these facades as "pettable", which means as much as appealing. Since the 16th century, houses in Moesia, partly in Thrace and in the surrounding Bulgarian regions, have been described as wooden by travelers [8].

Hans Dernschwam, a tradesman who was mainly active in Turkey and Slovakia at this time, stated in 1553 that the buildings in Sofia were made out of wood, the outer walls were decoupled by vertical wooden planks, and even the ceilings were covered by shingles. At this point, it should be mentioned that, according to several precise travel reports, half-timbered constructions with different infills and facade constructions were often referred to as wooden buildings. Even Dernschwam described the buildings in Plovdiv as being made out of wood but added below they were made out of tone, clay and wood, which indicates half-timbered buildings [9].

Besides Dernschwam, other travelers and diplomats mentioned as well that framework buildings in many Bulgarian regions were made with wood paneling. Henry Cavendish wrote in 1589, Silistra (Bulgaria) was quite a beautiful city, where one could see the building typology of low houses with wood paneling [10].

In the middle of the 17th century, Evliya Çelebi, an Ottoman traveler, determined within his journeys through Bulgarian regions that just in Moesia, on the Balkan and in regions of Shopi (Shopluk), houses were covered by planks. In Macedonia, in the White Sea regions, and in almost the whole of Thrace, buildings were massive houses or plastered framework buildings. About Lowetsch, he wrote

that the walls made out of pine were beautiful, and about Widin, Schumen and Warna he commented that the houses with wood paneling looked like engravings from the 19th century [11,12].

Already in the period between the 16th and the 17th century, centers with Black Sea Houses existed in the Bulgarian regions of Trjawna, Drjanowo, Malko Tarnowo and others, as well as in the eastern regions from Moesia, on the Balkan, and in the Strandzha Mountains. Without a doubt, these techniques have their origin in the Bulgarian tradition of wooden construction from early times, as block houses were built. The Black Sea House typology was transmitted by Bulgarian builders and craftsmen to Istanbul, where they reconstructed thousands of buildings, which were burned by multiple repeated fires [8].

The whole of traditional building architecture in the Balkan regions around the Strandzha Mountains, and until the 19th century, in the Rhodope Mountains, was wooden. The popular German building historian Professor Cornelius Gurlitt did 15 years of research on architecture in Istanbul and wrote in his publication that Bulgarians were the main builders there [13].

In the 19th century and at the beginning of the $20_{th}$ century, the Black Sea House was spread by Bulgarians over the regions around the Sea of Marmara and to the northeast coast of the Mediterranean Sea, as well as to certain points of the east side of the Black Sea. Though the best example of the occurrence of Black Sea House Typology in Turkey is still Istanbul, as it is the capital and center for wealthy people. Nowadays one can find Black Sea House typology widespread in smaller coastal towns in Bulgaria, especially in Warna, Nesebar, Sozopol, Tsarevo, Ahtopol, Malko Tarnovo, and in the Strandzha Mountains [8].

### 2.2.3. Requirement for Monument Preservation of Black Sea Houses

The monument-related reconstruction of Black Sea Houses requires the same restructuring measures as common half-timbered houses. Though the decisive factor for an appropriate material choice is the prevailing climate along the coast.

For example, Varna, a port at the Black Sea Coast of Bulgaria, showed through 2017 and 2018 an average humidity of about 74.5%, with a minimum of about 69.4% in August and a maximum of about 79% in December [14]. Additionally, in Sozopol, another city located further south along the Black Sea Coast, showed an average humidity of about 78% in 2016 [15]. Therefore, a very humid climate of humidity between approximately 70% and 80% over the whole year is prevailing in the coastal regions of Bulgaria. In addition, precipitation is high along the coast and the region is often affected by strong winds, especially during winter months. Moreover, high temperature differences prevail between summer and winter [16].

The TYPHABOARD fulfills the requirements for building materials in this region set by the prevailing climate conditions along the Black Sea Coast. Due to its high diffusion value and its excellent mildew resistance it can stand high humidity. And of course, the TYPHABOARD is perfectly suited for the energetic and functional retrofit of half-timber structures due to its properties for insulation as well as for bracing. Furthermore, moisture-induced expansion and shrinkage of the construction is provided through a joint of about 10 to 15 cm between the infill and the timber filled with swelling mortar, which is mixed with powdered typha material. This swelling mortar also serves as windproof sealing for cracks and holes [3].

Another important criterion within Black Sea House restoration is the preservation of their typical external wood paneling. This criterion is clearly fulfilled as TYPHABOARD is perfectly suitable as indoor insulation and therefore for installation from the inside.

### *2.3. Potentials and Experiences of TYPHABOARD Application*

### 2.3.1. First Application of TYPHABOARD in Monument Preservation

Within the renovation project of the half-timbered, medieval craftsman-made building in Pfeifergasse 9 in Nuremberg (see Figure 5), the application of TYPHABOARDS as insulation material

was tested for the first time. The property consisted of a front and back building and was located in the historic city center in the south of the town. Its origins date back to the 15th century and therefore it counts as an historical and architectural monument. The Altstadtfreunde associated purchased the property in 2003 for preservation and restoration, with the priority of safeguarding and sustainable reconstruction with maximum preservation of the original building substances. Due to the age of the building, structural modifications, and climate impacts over the past centuries, the property was identified as a seriously damaged historic monument (see Figure 6). The energetic reconstruction of the building had to be performed in adaption to the half-timbered structure, in accordance with the requirements of monument preservation, and with environmental compatibility as well as with the energy saving regulations of 2009.

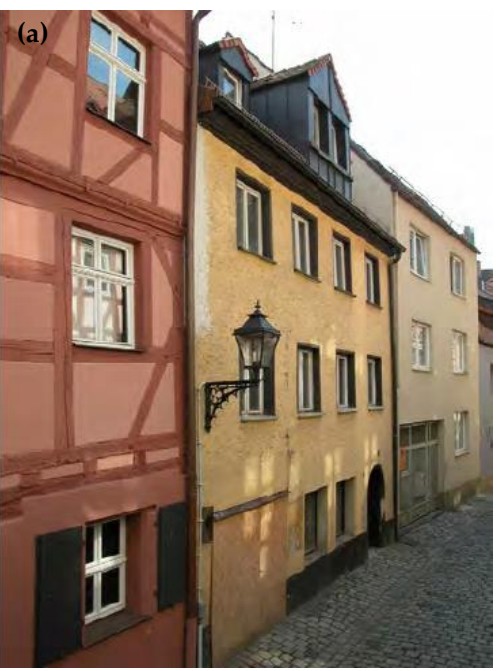 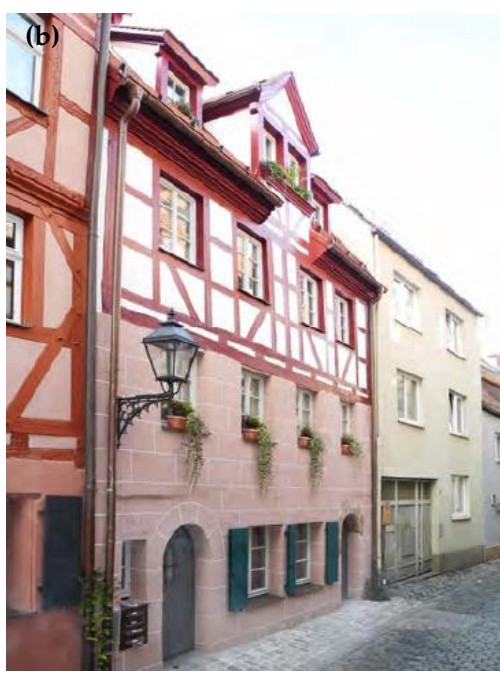

**Figure 5.** Building in Pfeifergasse 9 (**a**) before construction in 2003 (**b**) after construction in 2011, pictures: Alexandra Fritsch.

The building is characterized by small-tailored rooms, thin external walls, an asymmetrical structure with a tailcoat roof, severely deformed framework stories, and deficiencies in the bracing. The original building materials were sandstone, masonry, as well as wattle and daub masonry as infill, but solid bricks and various modern materials of reconstruction were also found (see Figure 6). The material requirements for the reconstruction were: the preservation of historic in-filling masonry; the necessary replacement of damaged masonry fillings; and using a material as an in-filling masonry that had insulation properties and would act as structural bracing to build a homogeneous external wall. A building material had to be found that enabled a diffusion-open internal thermal insulation system and the installation of wall heating, that was adjustable to the irregular framework construction as well as to the massive walls, and that was adequate for the framework fire-resisting work. TYPHABOARD was chosen as the building material for the renovation measures in Pfeifergasse 9 as it offers a variety of advantages and possibilities.

The restoration started in 2008 and required a feasibility study regarding the condition of the existing structure and the need of replacement of particular elements of it.. The first step was to restore and prepare the framework (see Figure 7a) by the installation of timber battens on the inner surface. Then, the TYPHABOARDS were cut with an approximately 10 to 15 mm wide joint all around between the size of the framework and the board. The boards were installed in the framework by screws and washers, and two boards were fixed one after another as one infill. A swelling mortar

mixed with powdered typha material was used for filling the connecting joints between the timber and the TYPHABOARD (see Figure 7b), as well as for cracks and holes in the timber framework. The addition of powdered typha material allowed for later swelling in case of further water penetration. The existing construction needed to be equalized and therefore was covered with a continuous internal board as far as possible without any air voids. To install wall heating, the pipes were directly screwed onto the boards (see Figure 8a) and the construction was then appropriately plastered with a clay plaster with typha seeds added (see Figure 8b).

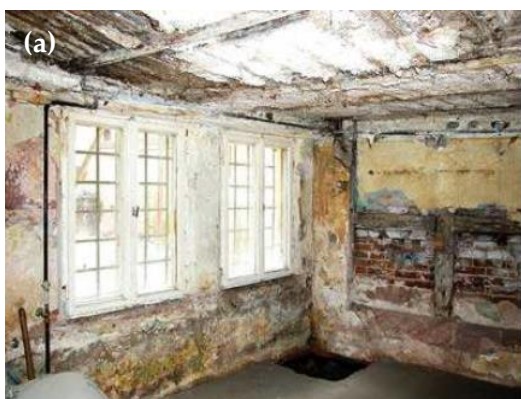 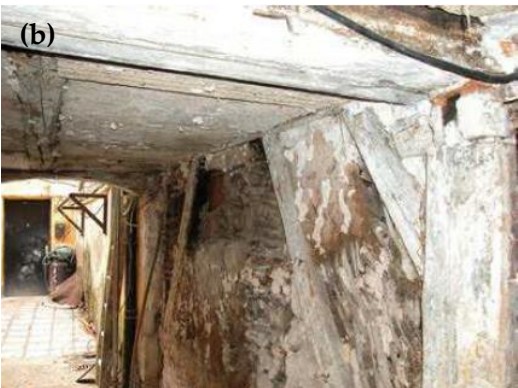

**Figure 6.** Damaged construction (**a**) original materials (**b**) with mixture of reconstruction material, pictures: Alexandra Fritsch.

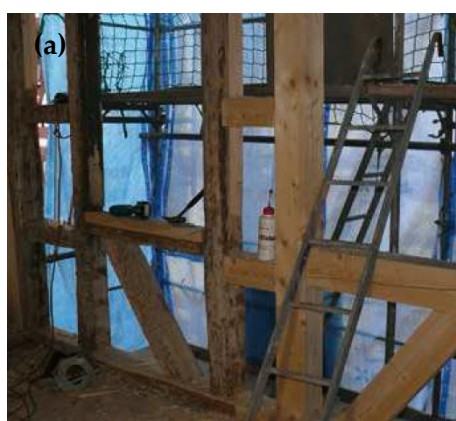 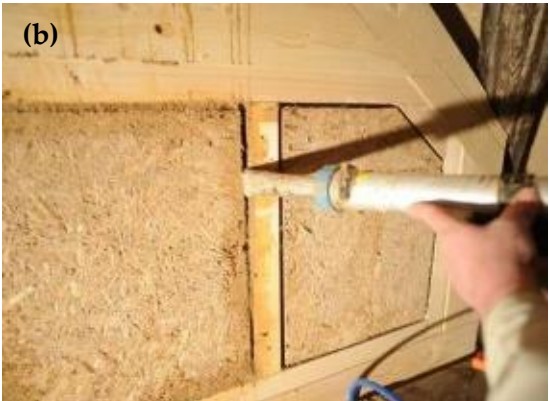

**Figure 7.** (**a**) Restoration of timber framework (**b**) TYPHABOARD infill and sealing of joints by typha swelling mortar, pictures: Fritsch Knodt Klug.

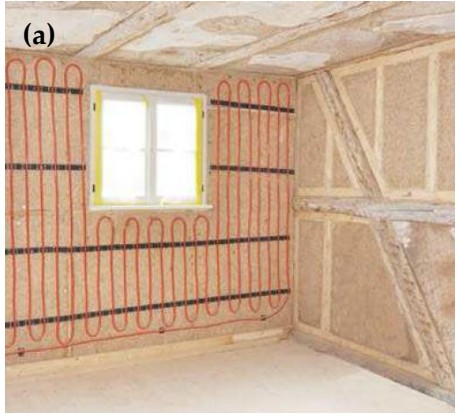 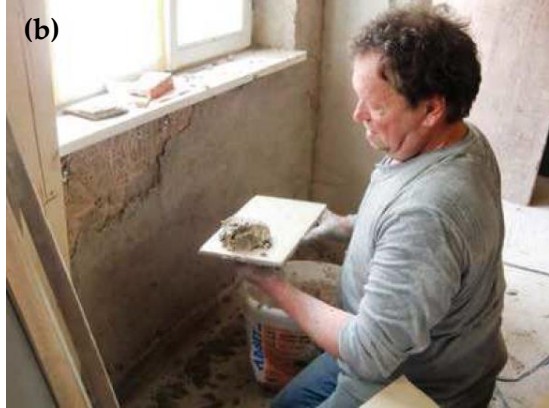

**Figure 8.** (**a**) Covered surface and screwing of the heating system, picture: Fritsch Knodt Klug (**b**) clay plastering on internal insulation, picture: Alexandra Fritsch.

For the whole monument reconstruction, approximately 300 m$^2$ of TYPHABOARD at a thickness of 40 to 60 mm was installed corresponding to approximately 15 m$^3$ of the material. Within this project, TYPHABOARD was tested the first time for practical use in framework restoration and proved to be perfectly suitable. It can be concluded that the TYPHABOARD could fulfill the initially required measures. The unique combination of insulation and load bearing, the simple workability, the good fire protection quality, the high mildew resistance and the perfect environmental performance convinced us of the qualities of the used material [3].

### 2.3.2. Simulation of Black Sea House Insulation with TYPHABOARD

Based on the experiences collected at the Nuremberg project, and based on TYPHABOARD, a study based on dynamic hygrothermal simulations was undertaken, in order to estimate the feasibility of the use of TYPHABOARD for the structural, functional and energy retrofitting and optimization of Black Sea Houses.

To calculate heat and moisture transport processes in the framework construction of a Black Sea House, a case study was implemented by the computer-aided hygrothermal dynamic simulation system, WUFI®-Pro. The program combines energetic and hygrothermal component calculations and provides temperature and humidity transport profiles of multilayered systems. The study was based on the climatic conditions of Varna (Bulgaria), a port city located at the Black Sea and therefore situated at just 14 m above sea level. Within the simulation, the behavior of a typical wall structure of a Black Sea House without and with additional TYPHABOARD internal insulation could be compared. Figure 9 shows the moisture situation of a wall structure with infill but without additional insulation at the surface of the interior plaster. This wall structure resulted in a relatively high heat transmission coefficient of 1.64 W/m$^2$K, but a surface dampness beneath 80%. Figure 10 shows the situation after an energetic renovation beneath the insulation layer with a back flow of 1 m$^3$/m$^2$h. The moisture analysis demonstrated a reduction of the heat transition coefficient of about 70% through an energetic restoration with the TYPHABOARD. This resulted in a significantly lower relative humidity beneath the insulating material and therefore there was no risk of moisture damage. Consequently, it can be concluded that the insulation with the TYPHABOARD is perfectly suited for a risk-free application on Black Sea Houses [4,7].

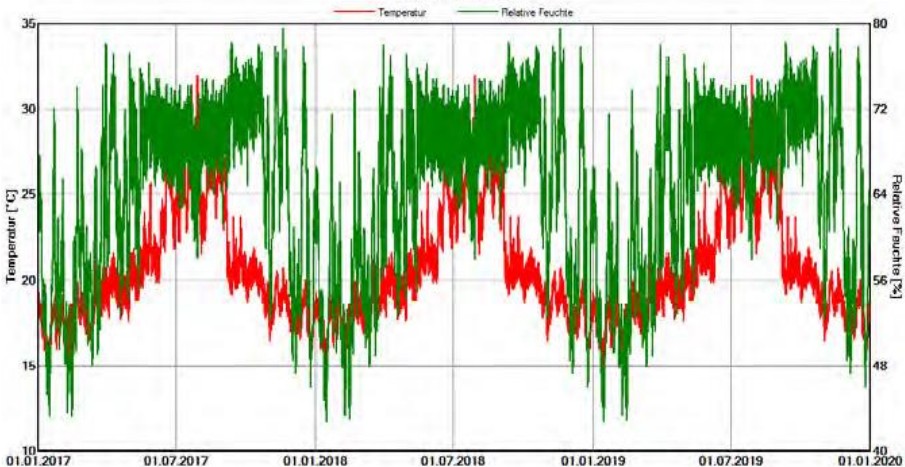

**Figure 9.** Temperature and moisture curve of the interior plaster of an unrenovated Black Sea House in Varna [7].

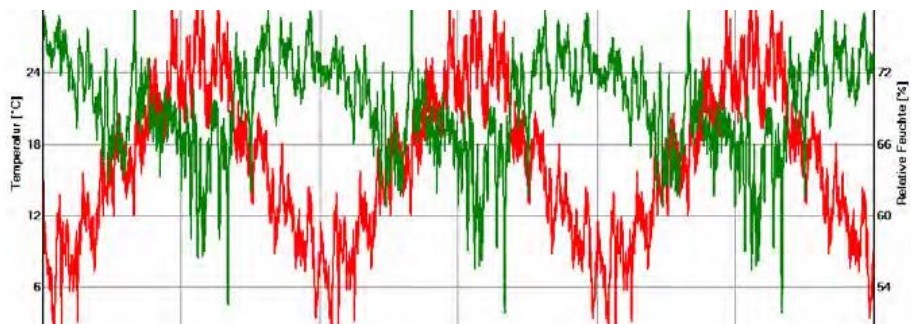

**Figure 10.** Temperature and moisture curve on the internal surface of an energetic renovated Black Sea House in Varna with a back flow of 1 m$^3$/m$^2$h [7].

## 3. Results

As a conclusion of both studies mentioned above—the 1:1 demonstration project in Nuremberg and the simulative study on the Black Sea house at the Black Sea coast in Bulgaria—it is safe to assume that TYPHABOARD is perfectly suitable for the monument preservation of Black Sea Houses. It is a qualified material to use as infill for half-timbered buildings as it has already been tested in Nuremberg and it can be optimally used as interior insulation as well as a compensation layer, but also as roof insulation. Furthermore, it can be easily installed and combined with historic building materials.

Due to its unique properties—high mechanical strength, good thermal insulation, capillary activity, processability—TYPHABOARD is a multi-talented material that fits the high requirements of historic building conservation in various climates.

The TYPHABOARD concept, especially the cultivation of typha plants, can be seen as an ecosystem service as the plant filters pollutants and moorlands fixes greenhouse gases. Due to the recyclability and biodegradability of the final product, TYPHABOARD is a particularly environmentally friendly building material. Moreover, the implementation of typha cultivation as well as TYPHABOARD production on site includes positive socio-economic effects as jobs can be generated and the regional economy in Bulgaria can be strengthened.

As it was assumed, as result of the DBU funded project on the feasibility of the TYPHABOARD concept in Bulgaria, there is a profound potential for raw material production, TYPHABOARD production and application at historic houses, and in Black Sea Houses in particular, locally and regionally.

On the other side, the simulation-based study and its results speak for a great opportunity for the architectonic and ethnographic typology of the Black Sea House to be saved and transferred to the next generation as a holistic historic artifact by applying TYPHABOARD for its physical and artisan renovation.

Based on the analysis of the potential for Black Sea House restoration in Bulgaria, further investigations need to be done through the real implementation on a selected reference object.

## 4. Discussion and Conclusions

In comparison to conventionally used types of insulation materials, the TYPHABOARD is more expensive (currently 1.000 €/m$^3$ in test production) as a material but in consideration of the entire renovation process it is certainly more affordable pricewise than other comparable construction systems for this purpose (for example clay, straw, various boards, vapor retarder layers, thermal insulation materials etc.,). Using TYPHABOARD—simultaneously a diffusion open construction and thermal insulation material, consisting of 50 mass % cattail and 50 mass % magnesite, and by this measure 100% compostable—for building renovation, less material layers are necessary than if other building systems were used. Moreover, the retrofit work with TYPHABOARD allows for simple and less time-consuming installation. Craftsmen are impressed by the easy processability of the material by

means of all usual tools. Therefore, the application of TYPHABOARD can prove its worth in the form of cost, time and effort saving.

Due to the simple manufacturing process of TYPHABOARD (as well as its being locally available as a raw material under natural conditions), and its uncomplicated installation on site, craftsmen without special expertise could be employed in Bulgaria. These advantages ensure a high potential for the implementation of the TYPHABOARD concept in Bulgaria in accordance with the support of sustainable and regional development, especially in structurally less developed regions.

Combining these advantages with the feasibility of TYPHABOARD as a universal retrofit for the Bulgarian Black Sea House, the concept deserves the attention of architects, civil engineers, spatial planners, politicians and the society in the region.

The project team is currently starting to undertake a knowledge and technology transfer action on the TYPHABOARD concept, in order to make it accessible to the relevant stakeholders' groups in Bulgaria and the region.

**Author Contributions:** The authors of this article contributed to it as follows: Conceptualization, G.G.; Methodology, G.G.; Software, G.G.; M.K. and C.L.; Validation, W.T. and M.K.; Formal Analysis, G.G.; Investigation, G.G. and C.L.; Resources, G.G.; Data, C.L. and G.G.; Writing-Original Draft Preparation, G.G. and C.L.; Writing-Review & Editing, C.L. and G.G.; Visualization, C.L.; Supervision, G.G.; Project Administration G.G.; Funding Acquisition, G.G.

**Funding:** This research received no external funding. There is references coming from three projects funded by the Deutsche Bundesstiftung Umwelt DBU, the German Federal Environmental Foundation in the past: —(AZ 10628), in the period from 1998 to 2001, focused on an exploration of the sustainable cultivation of typha in peat and moor areas in Germany and was conducted by the Technical University of Munich. Its final report was published under the German title "Rohrkolbenanbau in Niedermooren—Integration von Rohstoffgewinnung, Wasserreinigung zu einem nachhaltigen Nutzungskonzept" [2].—(AZ 27918), with the title "New Building Material for Environmentally Compatible and Structural Retrofitting in Monument Preservation", was conducted from 2010 to 2013 to test the application of TYPHABOARD as infill material and indoor insulation for the renovation of a half-timbered, medieval craftsman building in Nuremberg. Within this project, TYPHABOARD manufacturing technology was developed and the boards were installed as infill for damaged Fachwerk-fields, as well as indoor insulation at the Fachwerk façade of the pilot Fachwerk house [3].—(AZ 31996) on the TYPHABOARD concept, which ran from 2015 to 2017, investigated the implementation of TYPHABOARD as indoor insulation at the energy retrofit of multi-occupancy residential buildings with owner communities in Bulgaria. Its German title is "Prüfung der Übertragbarkeit eines neu entwickelten, innovativen, nachhaltigen Baustoffs zur thermischen Sanierung von Mehrfamilienhäusern mit Kleineigentümerstruktur in Bulgarien". Some of its contents and lessons learnt served as sources for the current publication.

**Acknowledgments:** The authors of this publication would like to thank all involved scientists, professionals, experts, political supporters and relevant institutional representatives for the supportive contributions on the way towards introducing the TYPHABOARD concept as a sustainable solution for the building construction and restoration process of the future.

**Conflicts of Interest:** The authors declare no conflict of interest.

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
