# Peer review of "TYPHABOARD in the Restoration of Historic Black Sea Houses in Bulgaria"

_sustainability, doi:10.3390/su11041000_

Round 1

Reviewer 1 Report

The paragraph 2.2.2 "History of Black Sea Houses" has incomplete references and it hasn't specific interest in this paper. A synthesis of the historical information could be added to the paragraph 2.2.1.

English language requires minor check (for instance: line 46 not "It's", but Its";  line 275  not "history", but "historical")

Author Response

Thank you for your helpful comments!

We still think, the place of the paragraph 2.2.2 is the right one.

The sources now are complete.

Corrections at line 46 and 275 are done now. It is more precise to say historic, and not historical.

Reviewer 2 Report

The paper, however very interesting in its premises, sounds still uncomplete.

While the introduction and section 2 (Material and method) is sufficient, referring to an oral source  (2.2, line 164) is inconsistent and unsuitable, at least it should be better justified! 

The the computer-aided hygrothermal dynamic simulation system WUFI® -Pro needs to be better described (line 320 ) also comparing the Black Sea House case with other case studies.

The results are  not supported by a description of quantitative or qualitative data on the compatibility of the described technology. It is necessary to focus better on the results of the research by now still insufficient (as stated in line 335 - 339).

The whole paper should be implemented with appropriate bibliographic research and references.

Author Response

Thank you for the useful comments!

In Building history often the information sources are oral ones.

The paper doesn't deal with the WUFI software as main topic, according to our understandinga further explination is not needed. The relevant experts' audience is aware of it.

The technology and its applicability are quite well described in the information sources, mentioned at the endof the paper. In the paper those are briefly, but exactly, described.

The bibliographic list was now updated.

Reviewer 3 Report

The article looks quite like a technical leaflet. Honestly, I do not see the relationship between the content and the title. There is no explanation why the authors have chosen completely different objacts as references. In what way can the authors compare the climatic conditions in Germany and Bulgaria? There is a reference to the history of traditional Bulgarian construction, but it proves almost nothing. Instead of describing the German case, a more detailed technical analysis of implementation in Bulgaria would be much more valuable.

Then, there is no reference to means and practice of the building conservation. How does the use of a typhaboard fit into the conservation doctrine? What technical solutions are adopted for retrofitting a structure of a building? How is the problem of wooden frames thermal bridges solved? Does the frame structure also apply to Bulgarian buildings?

On the sustainability and ecology: how do the authors explain the transport of raw material from Bulgaria or Romania, its processing in Germany and transport back to Bulgaria for use? In accordance to sustainability shouldn't it be a local material? Does it really simulate the Bulgarian economy to keep the raw material production leaving the processing for another countries? Is the industrial planting compatible with sustainability and pro-ecology? How about the other materials used in production process, are they also in accordance with the ecological regime?

Author Response

Good day,

Thank you for the comments!

Now we haveupdated some passages, according to your comments.

The Nuremberg case study was a physical case study, the one in Bulgaria is based on hygrothermal simulations.

The references regarding the structure and building technology of the Black Sea house describe well the details, which characterize the Black Sea House as a typology.

We have not mentioned anywhere, that someone would transport the raw material from Bulgaria, to produce TYPHABOARD in Germany, and transport TYPHABOARD back to Bulgaria. It would obviously speak against the entire content of the paper and the human logic. Now this is expressed clearer in the paper.

Round 2

Reviewer 2 Report

Comments to suggested revisions are not very convincing.

Research methods could be better explicated, with the support of more detailed sources .

Results could be better presented and discussed with the support of  quantitative or qualitative data. 

The discussion continues to be vague and generalized ( lines 352 - 362).

The integration to the bibliography was conducted by adding only 2 references. 

Author Response

Good day,

Thank you for the helpful critical comments!

We have now paid attention to all your comments from the 2. review round.

The research methods are now clearly described.

The argumentation regarding the TYPHABOARD feasibility for the application in the Black Sea Houses# retrofit is now more profound and supported by numbers.

The discussion is now better argumented.

The bibliography has been revised.

Of course, if one needs more detailed information, it can be found in our texts on the topic, to which we refer. These are based on our previous projects on TYPHABOARD. In a paper of this volume, we simply are not able to represent the entire information in its fillth, which we have collected and structured on hundreds of pages in our project reports and other related publications.

Attached you will find the revised version of the text.

Now we hope to have met you requirements and expectations.

Round 3

Reviewer 2 Report

The paper now returns an articulated research framework, both in the background and in the methodology. The theoretical approach and experimentation are described exhaustively and therefore may be of interest to the scientific community.

Author Response

Thank you!

We have revisedit according to your comments and are happy to be able to publish the article now.

Have a nice weekend!